# Humoral and Cellular Immune Responses to SARS-CoV-2 in Participants with Head and Neck Cancer

**DOI:** 10.3390/v17060848

**Published:** 2025-06-13

**Authors:** Luminita Mărutescu, Alexandru Enea, Nefeli-Maria Antoniadis, Marian Neculae, Diana Antonia Costea, Marcela Popa, Elena Dragu, Elena Codrici, Violeta Ristoiu, Bianca Galateanu, Ariana Hudita, Gratiela Gradisteanu Pircalabioru, Abdelali Filali-Mouhim, Serban Vifor Gabriel Bertesteanu, Veronica Lazăr, Carmen Chifiriuc, Raluca Grigore, Petronela Ancuta

**Affiliations:** 1Department of Botany and Microbiology, Faculty of Biology, University of Bucharest, 050663 Bucharest, Romania; luminita.marutescu@bio.unibuc.ro (L.M.); antoniadis.nefeli-maria22@s.bio.unibuc.ro (N.-M.A.); neculae.marian@s.bio.unibuc.ro (M.N.); costea.diana_antonia@s.bio.unibuc.ro (D.A.C.); luci.dragu@yahoo.com (E.D.); gratiela.gradisteanu@icub.unibuc.ro (G.G.P.); 2Research Institute of University of Bucharest, 050663 Bucharest, Romania; marcela.popa@bio.unibuc.ro (M.P.); violeta.ristoiu@bio.unibuc.ro (V.R.); ariana.hudita@bio.unibuc.ro (A.H.); 3ENT Department, Carol Davila University of Medicine and Pharmacy, 020021 Bucharest, Romania; alexandru-gabriel.enea@drd.umfcd.ro (A.E.); serban.bertesteanu@umfcd.ro (S.V.G.B.); 4Coltea Hospital, Rd District, 030171 Bucharest, Romania; 5Biochemistry-Proteomics Department, Victor Babes National Institute of Pathology, 050096 Bucharest, Romania; elena.codrici@umfcd.ro; 6Cell Biology and Histology Department, Carol Davila University of Medicine and Pharmacy, 050474 Bucharest, Romania; 7Department of Anatomy, Animal Physiology and Biophysics, Faculty of Biology, University of Bucharest, 050095 Bucharest, Romania; 8Department of Biochemistry and Molecular Biology, Faculty of Biology, University of Bucharest, 050095 Bucharest, Romania; 9Centre de Recherche du Centre Hospitalier de l’Université de Montréal, Montréal, QC H2X 0A9, Canada; alifilali2009@gmail.com; 10Département de Microbiologie, Infectiologie et Immunologie, Faculté de Médecine, Université de Montréal, Montréal, QC 6128, Canada

**Keywords:** SARS-CoV-2, antibodies, B-cells, T-cells, vaccine, correlates of immunity

## Abstract

Background: SARS-CoV-2 immunity is understudied in cancer patients. Here, we monitored natural/vaccine-induced SARS-CoV-2 immunity in patients with head and neck cancer (HNC) stratified as vaccinated (mRNA/adenovirus-based vaccines), convalescent, and hybrid immunity. Methods: Plasma/PBMC samples were collected from 49 patients with HNC and 14 non-oncologic controls recruited between August 2021 and March 2022. Longitudinal follow-up was performed on 25 HNC patients. Plasma antibodies (Abs) against Spike (S1/S2), receptor-binding domain (RBD), and nucleocapsid (NC) of IgG/IgA isotypes and 25 cytokines/chemokines were quantified using MILLIPLEX^®^ technology. The frequency, phenotype, and isotype of circulating SARS-CoV-2-specific B-cells were studied by flow cytometry using RBD tetramers (Tet^++^). The proliferation of B-cells and CD4+ and CD8+ T-cells in response to Spike/NC peptides was monitored by a carboxyfluorescein succinimidyl ester (CFSE) assay. Results: Plasma SARS-CoV-2 S1/S2/RBD IgG/IgA Abs were detected in all HNC participants at enrollment median time since immunization (TSI) 117 days at levels similar to controls and were significantly higher in convalescent/hybrid versus vaccinated. NC IgG/IgA Abs were only detected after infection. The frequency of Tet++ B-cells, enriched in the CD27+ memory phenotype and IgG/IgA isotype, positively correlated with plasma levels of RBD IgG/IgA Abs and Spike-specific CD4+ T-cell proliferation, regardless of the immunization status and TSI. Spike/NC-specific B-cell proliferation reached the highest levels in convalescent HNC and was positively correlated with NC IgG Abs, but not with the frequency of Tet++ B-cells. Finally, Tet++ B-cell frequencies remained stable between the two subsequent visits (median TSI: 117 versus 341 days), indicating their ability to persist for a relatively long time. Conclusions: This study monitored SARS-CoV-2 humoral/cellular immunity in an HNC cohort relative to non-oncologic participants and demonstrates that SARS-CoV-2-specific B-cells persist beyond 11 months post-immunization. These findings have implications for the management of HNC in the context of SARS-CoV-2 infection and other viral infections.

## 1. Introduction

Severe acute respiratory syndrome coronavirus 2 (SARS-CoV-2) emerged in December 2019 in the Wuhan region of China and is the etiological agent of the novel coronavirus disease 2019 (COVID-19) [1]. As of October 2023, the Center for Systems Science and Engineering at Johns Hopkins University (https://coronavirus.jhu.edu/map.html, accessed on October 3rd, 2023) reported that SARS-CoV-2 has infected over 676 million people, resulting in over 6.8 million reported deaths, with over 13 billion doses of SARS-CoV-2 vaccines administered worldwide.

SARS-CoV-2 enters target cells (e.g., epithelial cells in the respiratory tract) via angiotensin-converting enzyme 2 (ACE2), which serves as the primary entry receptor [2]. The secondary entry receptor transmembrane protease serine 2 (TMPRSS2) is essential for spike envelope glycoprotein cleavage [2,3,4,5]. ACE2 and TMPRSS2 are co-expressed at high levels in the nasal epithelium, whereas epithelial cells from the small airway mainly express TMPRSS2 [6]. This differential expression of SARS-CoV-2 receptors in the upper versus lower respiratory tract points to anatomical site-specific particularities in the mechanisms of viral entry [5,7]. The pathogenicity of SARS-CoV-2 depends on its ability to transition from the upper (nasal) to the lower respiratory tract (pulmonary) compartment [8,9].

The new generation of mRNA-based vaccines has allowed for the rapid implementation of SARS-CoV-2 vaccination worldwide, starting in early 2021 [5]. Other SARS-CoV-2 vaccines, such as adenovirus vector-based, inactivated, and recombinant protein vaccines, have been used in parallel [5]. Large-scale immune monitoring studies have been performed to demonstrate the efficacy of SARS-CoV-2 vaccines in promoting robust humoral/cellular immunity [8,9]. Tools have been developed to distinguish between immunization induced by vaccination and natural infection, using the concomitant measurement of Spike (S1, S2, receptor-binding domain (RBD) Abs induced by vaccination and natural infection) and nucleocapsid (NC) Abs, induced only by natural infection [10,11,12,13]. Early studies demonstrated the superior ability of IgA compared with IgG Abs to neutralize SARS-CoV-2 virions [14] and supported the importance of IgA Abs for efficient immune protection at the respiratory mucosal barriers [8,9].

The severity of COVID-19 symptoms is associated with age, sex, metabolic deficiencies, and comorbidities [15,16], as well as host genetic factors [17]. Both innate and adaptive immunity limit SARS-CoV-2 disease severity [8,9,12,13], and mucosal barrier innate immunity is essential to prevent viral spread from the upper respiratory tract to the lungs [18,19]. SARS-CoV-2 natural immunity and vaccine efficacy is compromised in high-risk groups, including cancer [20,21,22,23]. Although cancer patients exhibit altered immune responses and are considered at risk for COVID-19 [24,25,26,27], long-term humoral and cellular immune responses against SARS-CoV-2 have not been extensively studied in different types of malignancies [23,28], such as head and neck cancer (HNC) [29]. In 2020, HNC was the seventh most common cancer worldwide [30], predominantly in males, with smoking being the main risk factor [31,32,33,34]. HNC is highly prevalent in countries such as Romania [35,36,37]. Humoral and cellular immunity against SARS-CoV-2, induced by vaccination and/or natural infection, is understudied in this specific group, which is at high risk of acquiring viral infections due to immune suppression subsequent to heavy oncologic treatments [28,38,39].

To address this knowledge gap, we performed a comprehensive immune monitoring study to document the quality and duration of natural and vaccine-induced humoral and cellular immune responses to SARS-CoV-2 in a cohort of patients with HNC (n = 49), with a fraction receiving oncologic treatments (surgery, radiotherapy, and/or chemotherapy) at Coltea Hospital, Bucharest, Romania. A control group without oncological conditions (n = 14) was recruited in parallel. Multiplex bead approaches were used for the simultaneous quantification of eight distinct types of SARS-CoV-2 Abs, i.e., spike (S1, S2, and RBD) and nucleocapsid (NC) Abs of IgG and IgA isotypes, as well as 25 cytokines in plasma samples. We also used polychromatic flow cytometry on peripheral blood mononuclear cells (PBMCs) to explore, ex vivo, the frequency of SARS-CoV-2-specific B-cells and binding RBD tetramers (Tet^++^) and to measure the proliferation potential of B-cells and T-cells in response to SARS-CoV-2 Spike and NC proteins. Finally, a longitudinal follow-up was performed in a group of HNC participants (n = 25) at two subsequent visits at median times since immunization (TSI) of 117 and 341 days. Our results provide novel insights into the quality and duration of natural and/or vaccine-induced immunity against SARS-CoV-2 in patients with HNC and are deposited on BioRxiv [40]. This knowledge opens new questions regarding the molecular mechanisms underlying antiviral mucosal immunity at the respiratory barrier in patients with HNC. In addition, our study has important implications for the management of patients with HNC in the context of pandemic airborne viral infections. These findings are also relevant in the context of the most recent clinical trials conducted by Moderna on the use of personalized mRNA vaccines for the treatment of solid cancers [41].

## 2. Materials and Methods

### 2.1. Study Cohort

The study was performed on patients with HNC (n = 49) and the control (CTR) group (n = 14; without neoplasm history, who presented to the hospital for non-oncologic checkup (n = 7) or were healthy health care volunteers (n = 7)). Participants were recruited at the Coltea Hospital, Bucharest, Romania, between August 2021 and March 2022 (Appendix A). The descriptions of the study participants are detailed in Table 1 in terms of sex, age, body mass index (BMI), SARS-CoV-2 infection symptoms, immunization status (vaccination and/or natural infection), time since immunization (TSI), diabetes, and oncologic pathology details. Peripheral blood samples were collected from the participants at one visit (for all participants) or two subsequent visits (n = 25 HNC and n = 3 CTR participants). Plasma and peripheral blood mononuclear cells (PBMCs) were isolated from whole blood and stored at −80 °C until use.

### 2.2. Plasma and PBMC Sample Separation

Peripheral blood (10–40 mL blood per participant) was collected by venipuncture in EDTA tubes (BD vacutainer; Plymouth, UK) and processed within two hours. The blood was centrifuged at 400× *g* for 10 min at room temperature. The superior plasma fraction was collected, dispatched in 5–10 aliquots in 1 mL, and stored at −80 °C until use. PBMCs were isolated from whole blood using Ficoll–Paque (GE17-1440-02 Ficoll^®^ Paque Plus; Uppsala, Sweden) density gradient centrifugation at 400× *g* for 30 min at room temperature. PBMCs were cryopreserved in fetal bovine serum (FBS, 2394341RP; Gibco, Mexico) containing 10% DMSO (D4540-1L; Sigma, Burlington, MA, USA) at −80 °C until further use, as previously described [42].

### 2.3. Luminex Detection of Plasma IgG and IgA Abs to SARS-CoV-2

Specific IgG and IgA Ab profiles against four SARS-CoV-2 viral antigens (receptor-binding domain (RBD), spike 1 subunit (S1), and spike 2 subunit (S2) of spike protein, and nucleocapsid (NC)) were monitored in the plasma samples using Millipore’s MILLIPLEX^®^ MAP SARS-CoV-2 Antigen EMD panel IgG and IgA (Merck, Darmstadt, Germany), according to the manufacturer’s instructions. Briefly, 1/100 diluted plasma samples were incubated with fluorescent magnetic beads coated with the following viral recombinant SARS-CoV-2 proteins: Spike S1, Spike S2, RBD, and NC. Then, PE-conjugated anti-human IgG or anti-human IgA was added to detect beads coated with plasma Abs. The antigen–Ab complexes on the beads were analyzed using a Luminex^®^ 200™ system (Luminex Corp., Austin, TX, USA) to generate the median fluorescence intensity (MFI) of the signal per 100 beads. Data acquisition and analysis were performed using the xPONENT 4.2 software (Luminex, Luminex Corp., Austin, TX, USA), and the calibration curves were generated with a 5-parameter logistic fit. The results were expressed as relative fluorescence units (RFUs).

### 2.4. Flow Cytometry Quantification of SARS-CoV-2 RBD-Specific B-Cells in PMBCs

SARS-CoV-2-specific B-cells present in PBMC samples were detected using the SARS-CoV-2 RBD B-cell analysis kit (Miltenyi Biotec, Bergisch Gladbach, Germany), according to the manufacturer’s protocol. B-cell tetramer staining technology was developed to address the need for high-throughput strategies to monitor circulating SARS-CoV-2-specific B-cells, which represent a likely source of SARS-CoV-2 Abs, as previously reported [43,44,45,46,47,48]. Briefly, 10^7^ PBMCs were stained with two solutions of SARS-CoV-2 Spike RBD tetramers (Tet) conjugated with PE and PE-Vio770. Tetramers were generated by incubating biotin-labeled recombinant SARS-CoV-2 Spike RBD protein with streptavidin-PE and streptavidin-PE-Vio770. Concomitant surface staining was performed with fluorochrome-conjugated Abs: CD19 APC-Vio770 Abs (clone LT19, isotype: mouse IgG1k) and CD27 VioBright FITC Abs (clone M-T271, isotype: mouse IgG1k) and/or with the anti-isotype Abs VioBlue-conjugated anti-IgG (clone IS11-3B2.2.3, isotype: mouse IgG1k), VioGreen-conjugated anti-IgA (clone IS11-8E10, isotype: mouse IgG1k), and APC-conjugated anti-IgM clone (PJ2-22H3, isotype: mouse IgG1). After incubation at room temperature for 30 min, the PBMCs were washed with PEB buffer (PBS, pH 7.2, 0.5% bovine serum albumin (#SLCF3210 Merck-Sigma, Durham, USA), and 2 mM EDTA (Merck-Sigma, Durham, USA)) and analyzed by flow cytometry using a 3-laser Gallios flow cytometer (Beckman Coulter, Brea, USA). Cell debris, doublets, and dead cells were excluded from the analysis based on the size/granularity scatter signals. CD19^+^ B-cells with and without a memory (CD27^+^) phenotype were analyzed for the presence of RBD-specific B-cells identified using the double discrimination method of tetramer-positive (Tet^++^) cells (PE/PE-Vio770). The IgM, IgG, and IgA isotypes of the total and Tet^++^ B-cells were also analyzed. The fluorescence minus one (FMO) strategy was used to establish positivity gates, as previously reported [49].

### 2.5. Carboxyfluorescein Succinimidyl Ester (CFSE) Proliferation Assay

The CFSE proliferation assay was performed, as previously reported [50], to monitor SARS-CoV-2-specific proliferation of immune cells from PBMCs of HNC and NHNC study participants. Briefly, PBMCs (2 × 10^6^ cells/mL) were stained with 0.5 μM CFSE (BioLegend, San Diego, CA, USA; B377681) in PBS 1X (Gibco, Geel, Belgium; 18912-014), for 8 min at 37 °C and protected from light. After washing, the cells were suspended in proliferation media (RPMI 1640, L-glutamine and HEPES (Gibco, New York, USA; A10491-01) supplemented with 1% penicillin/streptomycin (Gibco; New York, USA; 15140122) and 10% human serum (ZenBio, Durham, USA; HSER-ABP, SER052318). After inoculation at a density of 0.25 × 10^6^ cells/well in 96-well microtiter plates (TPP; Saint Louis, USA; 96F92696) at a final volume of 250 µL/well, cells were stimulated with SARS-CoV-2 peptide pools (PepTivator^®^ SARS-CoV-2 Prot_N 130-126-699; SARS-CoV-2 Prot_S 130127953) (0.6 nmol/mL each) (Miltenyi Biotec; Bergisch Gladbach, Germany). After 6 days of incubation at 37 °C and 5% CO_2_, the cells were harvested and labeled on the surface with the following monoclonal Abs from BioLegend (San Diego, USA): APC-conjugated anti-CD3 Abs (clone Mouse IgGak OKT3), PE-conjugated anti-CD4 (clone Mouse IgG1k RPA-T4), PerCP-conjugated anti-CD8 (clone Mouse IgG1k SK1), APC-conjugated anti-CD19 (clone HIB19), and PE-conjugated anti-CD56 (clone HCD56). Following staining, cells were washed twice with FACS buffer (PBS 1X, 0.02% NaN3, 10% FBS) and analyzed using a BD Accuri C6 Plus flow cytometer (BD Biosciences, New Jersey, USA). Data analysis was performed using BD Accuri C6 software (BD Biosciences, New Jersey, USA).

### 2.6. Luminex Quantification of Plasma Th17 Cytokine Profiles

Plasma cytokine and chemokine levels were measured using the Human Th17 Magnetic Bead Panel (TH17MAG-14kMILLIPLEX MAP, Merck, Darmstadt, Germany), allowing simultaneous quantification of the following cytokines: IL-1β, IL-2, IL-4, IL-5, IL-6, IL-9, IL-10, IL-12p70, IL-13, IL-15, IL-17A, IL-17E/IL-25, IL-17F, IL-21, IL-22, IL-23, IL-27, IL-28A, IL-31, IL-33, GM-CSF, IFN-γ, MIP-3α/CCL20, TNF-α, and TNF-β. The assay was performed according to the manufacturer’s instructions. Briefly, 25 µL plasma samples were incubated with fluorescent magnetic beads coated with Abs against cytokines. PE-conjugated anti-cytokine Abs were then added to detect beads coated with cytokines. The antigen–Ab complexes on the beads were analyzed using a Luminex^®^ 200™ system (Luminex, Luminex Corp., Austin, TX, USA) to generate the median fluorescence intensity (MFI) of the signal per 100 beads. The use of a standard curve for each cytokine allowed the expression of the results in pg/mL.

### 2.7. Statistical Analysis

For comparisons between groups and correlations, statistical analyses were performed using Prism 7 software (GraphPad, Inc., La Joya, CA, USA). Statistical tests were applied to determine statistical significance between matched groups (Wilcoxon test) and unmatched comparisons between multiple groups (Kruskal–Wallis test), with the *p*-values indicated in the graphs and explained in the figure legends. Additionally, the Spearman correlation model was applied to identify predictors of immunological readouts, with *p-* and r-values indicated in the graphs and explained in the figure legends.

#### Linear Regression

To investigate the association between humoral and cellular immune response parameters that predict specific SARS-CoV-2 immunity outcomes (i.e., SARS-CoV-2 RBD IgG and IgA Abs and the frequencies of circulating SARS-CoV-2 Spike RBD-specific B-cells (Tet^++^ B-cells)), a linear regression model was implemented with adjustments for numerical variables (i.e., age, BMI, and TSI) and/or categorical variables (i.e., sex, smoking, alcohol, toxic environment, and diabetes). The regression coefficients and *p*-values for each immune parameter were calculated (Appendix A). *p*-value adjustment for multiple testing hypothesis was performed according to the method of Benjamin and Hochberg [51], which controls the false discovery rate (FDR) with adjusted *p*-value cut-offs of 0.05. Linear regression analyses were performed using the statistical package R [52] version 4.2.3, with the R pwr package being used for sample size post hoc analysis.

### 2.8. SARS-CoV-2 Variants Circulating in Romania at the Time of the Study

In Romania, the Wuhan strain remained dominant through the end of 2020. However, beginning in the second half of February 2021, it was rapidly replaced by the Alpha variant. During the spring of 2021, isolated cases of the Beta and Gamma variants were detected, though these did not reach dominance. The Delta variant began circulating in late April 2021 and became the predominant strain by July 2021. Toward the end of 2021, the Omicron variant emerged and dominated throughout most of 2022, although by late that year, it began to co-circulate with other non-Omicron variants [53].

## 3. Results

### 3.1. Study Participant Clinical Characteristics

Table 1 and Appendix A describe the study cohort. The HNC and control groups differed in terms of age (median 65 and 56 years old, respectively; Mann–Whitney *p* = 0.019) and male/female ratios (Fisher’s exact test *p* = 0.0002), with HNC including more males than females and being older than controls. The HNC versus control groups were similar in terms of immunization status (i.e., vaccinated (n = 21), convalescent (n = 13), hybrid immunity (n = 15) and symptoms (i.e., mild to moderate, Fisher’s test)) upon natural SARS-CoV-2 infection, as well as TSI and diabetes prevalence, with a tendency for decreased BMI in HNC compared with controls (Mann–Whitney *p* = 0.0696) (Table 1). Vaccinated HNC participants (20/49) received BioNTech/Pfizer BNT162b2 (10/20), Johnson & Johnson/Janssen Ad26.COV.2.S (4/20), Moderna mRNA-1273 (4/20), or Oxford–AstraZeneca vaccines (ChAdOx1 CoV-19) (1/20), while all vaccinated controls (6/14) received the BioNTech/Pfizer BNT162b2 vaccine (Appendix A). The oncologic pathology in the HNC group included rhinopharyngeal, oropharyngeal, mandibular, laryngeal, pelvilingual, parotidian, sinus, and thyroid neoplasms, as well as non-Hodkin lymphoma and unknown primary cancers (Table 1; Appendix A). A major fraction of HNC patients underwent recurrent surgical tumor excision (35/49) and/or received oncologic treatment (16/49 received radiotherapy and 11/49 received radiotherapy in combination with chemotherapy (cisplatin, carboplatin, paclitaxel) and/or immunotherapy (Nivolumab, Pembrolizumab)), while 21/49 patients were naïve to any oncologic treatment at the time of blood collection (Appendix A). Within the HNC group, there were no statistical differences in terms of age, BMI, and TSI between males (n = 43) and females (n = 6) (Appendix A), nor between vaccinated (n = 21), convalescent (n = 13), and hybrid immunity (n = 15) participants (Appendix A).

### 3.2. Plasma Levels of SARS-CoV-2 IgG and IgA Abs in HNC Patients

Analysis of plasma samples collected at the first visit (median TSI 107 days for HNC and 65 days for controls (Table 1) showed detectable levels of SARS-CoV-2 IgG and IgA Abs (S1, S2, RBD, and NC) in the majority of participants (Figure 1; Appendix A).

Concerning the SARS-CoV-2 Abs of IgG isotype, HNC and control participants exhibited similar levels of plasma S2 and NC IgG Abs, while S1 and RBD IgG Abs titers tended to be lower in HNC versus control participants (Figure 1A). Among HNC participants, S1, S2, and RBD IgG Ab levels were significantly higher in convalescent and/or hybrid immunity than in vaccinated participants, with no significant differences between convalescent and hybrid immunity participants (Figure 1B). As expected, NC IgG Abs were detected only in convalescent and hybrid immunity HNC participants (Figure 1B). Spearman models revealed no statistically significant correlations between S1, S2, RBD, or NC IgG Ab titers and age, TSI, or BMI. Regarding SARS-CoV-2 Abs of IgA isotypes, their levels were overall lower when compared with IgG Abs levels but were statistically similar between HNC and controls (Figure 1A,C).

Among HNC participants, convalescent versus vaccinated status was associated with significantly higher levels of S1, S2, and RBD IgA Abs, with no statistically significant differences between convalescent and hybrid immunity statuses (Figure 1D). While age was not a predictor of IgA Ab levels, Spearman correlation models identified TSI as a negative predictor of plasma NC (*p* = 0.0356, r = −0.3009), but not S1, S2, or RBD IgA Ab titers. Finally, BMI was positively correlated with RBD IgA Abs (*p* = 0.0391; r = 3121) but not with S1, S2, and NC IgA Ab levels.

Together, these results reveal the induction of effective SARS-CoV-2 IgG and IgA Ab responses by the HNC participants at levels similar to controls, with the highest Abs titers in convalescent and hybrid immunity compared with vaccinated participants, regardless of age, TSI, and BMI, with the exception of NC IgA Abs levels that decreased with TSI, and RBD IgA Ab levels were positively correlated with BMI.

### 3.3. Circulating SARS-CoV-2-Specific B-Cells in HNC Participants

To further characterize SARS-CoV-2 humoral immunity, circulating total B-cells (CD19^+^) were analyzed for their SARS-CoV-2 specificity and memory phenotype (CD27^+^) by flow cytometry upon staining with SARS-CoV-2 Spike RBD tetramers (Tet^++^) and CD27 Abs, respectively, using the gating strategy depicted in Figure 2A. This is a commercially available SARS-CoV-2 RBD B-cell analysis kit (Miltenyi Biotec) developed since 2020 and has been used in multiple SARS-CoV-2 immune monitoring studies [43,44,45,46,47,48]. The total (CD19^+^) and memory (CD19^+^CD27^+^) Tet^++^ B-cells were present at various frequencies in the peripheral blood of HNC participants, with no statistically significant differences compared with control participants (Figure 2B). Within the HNC group, Tet^++^ B-cells were slightly enriched in the memory phenotype compared with total B-cells (median 29.3% versus 23.8% CD27^+^ B-cells, *p* = 0.0272; Figure 2C). No statistical differences were observed in the frequencies of total and memory Tet^++^ B-cells among the vaccinated, convalescent, and hybrid immunity groups (Figure 2D). The frequency of Tet^++^ B-cells was positively correlated with the plasma levels of SARS-CoV-2 RBD Abs of both IgG and IgA isotypes (Figure 2E). The % of Tet^++^ B-cells did not correlate with TSI (Figure 2F), indicating the stability of SARS-CoV-2 B-cell memory after infection/vaccination, at least for the TSI studied.

In parallel, we investigated the IgM, IgG, and IgA isotypes of Tet^++^ B-cells using the gating strategy depicted in Figure 3A. Among HNC participants, Tet^++^ B-cells were distinguished from total B-cells by lower expression of IgM and higher expression of IgG and IgA (Figure 3B). This suggests a more effective immunoglobulin class switch from IgM to IgG or IgA following exposure to SARS-CoV-2. The isotype profile of Tet^++^ B-cells was similar in HNC and control participants (Figure 3C) and did not vary in HNC participants stratified based on immunization status (i.e., vaccinated, convalescent, and hybrid immunity groups) (Figure 3D).

Together, these results revealed the presence of RBD-specific B-cells, with a CD27+ memory phenotype and IgM/IgG/IgA isotype in HNC participants, at frequencies similar to controls, irrespective of immunization status or TSI. The positive correlation between the frequency of RBD-specific B-cells and the plasma levels of RBD-specific IgG/IgA Abs suggests that circulating Tet^++^ B-cells may represent a source of neutralizing Abs against SARS-CoV-2. Thus, RBD-specific B-cells represent an important SARS-CoV-2 immunity outcome to monitor in immune surveillance studies.

### 3.4. Proliferation of SARS-CoV-2-Specific B-Cells

In addition to the use of tetramers for the identification of SARS-CoV-2-specific B-cells ex vivo, the CFSE dilution assay was used in parallel to identify B-cells proliferating in response to SARS-CoV-2 proteins in vitro, as a readout of their survival capacity in vivo. The results revealed the proliferation (CFSE^low^) of CD19^+^ B-cells upon cultivation of PBMC in the presence or absence of SARS-CoV-2 Spike and NC recombinant proteins (Figure 4A,B). Statistical analysis showed a significantly higher proliferation of CD19^+^ B-cells upon exposure to SARS-CoV-2 Spike compared with the NC protein (Figure 4C), consistent with the well-described superior immunogenicity of Spike relative to NC [12,13,54]. The proliferation of B-cells in response to SARS-CoV-2 Spike and NC showed statistically significant differences among the HNC group stratified based on immunization status, with the highest levels of Spike- and NC-specific B-cell proliferation in convalescent versus vaccinated and hybrid immunity participants (Figure 4D).

The Spearman correlation model was applied to analyze the relationship between the frequency/isotype of circulating Tet^++^ B-cells and the plasma levels of SARS-CoV-2 Abs in HNC participants. The frequency of CD19^+^ B-cells proliferation in response to Spike and NC proteins did not correlate with the frequency of Tet^++^ B-cells (Figure 4E), suggesting that some circulating Tet^++^ B-cells may have an impaired survival capacity. Nevertheless, consistent with the predominant B-cell proliferation in convalescent HNC participants (Figure 4D), the frequency of proliferating SARS-CoV-2-specific B-cells was positively correlated with plasma levels of NC IgG Abs, with a significant correlation for Spike-specific B-cells (Figure 4F, left panel) and a marginally significant correlation for NC-specific B-cells (Figure 4F, right panel). Thus, B-cell proliferation in response to SARS-CoV-2 peptides was the most robust in convalescent HNC participants and positively correlated with plasma levels of NC-specific IgG Abs.

### 3.5. Proliferation of SARS-CoV-2-Specific CD4+ and CD8+ T-Cells

The contribution of CD4^+^ and CD8^+^ T-cells to SARS-CoV-2 immunity is well recognized in the general population [8,9]. To identify the cellular components of SARS-CoV-2 antiviral responses among HNC participants, we investigated the proliferation of CD4^+^ and CD8^+^ T-cells in response to recombinant SARS-CoV-2 Spike and NC proteins (Appendix A). Similar to the results on B-cell proliferation (Figure 4C), Spike showed superior immunogenicity compared with NC peptides for the induction of CD4^+^ T-cells and CD8^+^ T-cells proliferation (Appendix A). The % of proliferating (CFSE^low^) CD4^+^ and CD8^+^ T-cells in response to Spike and NC peptides was similar in HNC participants regardless of immunization status (Appendix A). The frequency of CD4^+^ T-cell proliferation in response to Spike, but not NC peptides, was positively correlated with the frequency of circulating Tet++ B-cells (Appendix A), consistent with the well-documented cross-talk between CD4^+^ T-cells and B-cells in sustaining humoral responses [55]. Such correlations were not observed between the frequency of proliferating Spike/NC-specific CD8^+^ T-cells and circulating Tet^++^ B-cells.

### 3.6. Systemic Cytokines Levels

Plasma levels of proinflammatory cytokines have been reported to be elevated in patients with severe COVID-19 [56]. To evaluate the systemic immune activation in relation to the above-mentioned monitored humoral and cellular SARS-CoV-2 immunity readouts, plasma levels of 25 cytokines were quantified using the Luminex^®^xMAP^®^ technology, using low- (i.e., IL-17F, GM-CSF, IL-10, IL-22, IL-4, IL-23, IL17E/IL-25, IL-27, IL-31, TNF-β, IL-28A) and (i.e., IFN-γ, CCL20, IL-12p70, IL-13, IL-15, IL-17A, IL-9, IL-1*β*, IL-33, IL-2, IL-21, IL-5, IL-6, TNF-α) high-sensitivity standard curves, proportional to the physiological concentrations of specific cytokines in human plasma (Appendix A). Cytokines present at detectable levels in the plasma were further grouped into the Th1/Th2/Th17 lineage and inflammatory cytokines (Appendix A). Plasma levels of these cytokines tended to be increased in HNC compared with control participants, with statistically significant differences being observed for CCL20 (*p* < 0.0001) and IL-33 (*p* = 0.0011) and marginally significant differences for IL-21 (*p* = 0.0597) (Appendix A). Stratification based on immunization status did not reveal any statistically significant differences between the groups for these three cytokines (Appendix A). Finally, a Spearman’s correlation model was used to identify cytokines that could be used as predictors of major immune outcomes. Of particular importance, although plasma levels of IL-6 did not distinguish HNC from control participants (Appendix A), among HNC participants, IL-6 levels negatively correlated with plasma levels of RBD IgG (*p* = 0.0193, r = −0.3333) and IgA (*p* = 0.0252, r = −0.3196) Abs as well as with the frequencies of Tet^++^ B-cells (*p* = 0.0024, r = −0.4233) (Appendix A).

### 3.7. Linear Regression to Identify Correlates of SARS-CoV-2-Specific Humoral Immunity Outcomes

To identify predictors of the three major humoral immunity outcomes (i.e., RBD IgG and RBD IgA Abs, and Tet^++^ B-cells) among the panel of immunological measurements realized in this study, a linear regression model was used. The results, including the regression coefficient, *p*-values, and adjusted *p*-values (Adj. p), are presented in Appendix A. A negative/positive coefficient indicated a negative or positive correlation. In addition to the crude model, adjustments for numerical (i.e., age, BMI, and TSI) and/or categorical (i.e., sex, smoking, alcohol, toxic environment, and diabetes) variables, as well as adjustments for TSI alone, were performed (Appendix A).

For SARS-CoV-2 Abs measurements (Figure 1), levels of S1, S2, and NC IgG Abs were positively correlated with RBD IgG Ab levels, while levels of S1 and S2 IgG and IgA Ab levels were positively correlated with RBD IgA Abs levels, with statistical significance (Adj. *p* < 0.05) being maintained upon adjustment for numerical and categorical variables (Appendix A). For B-cell measurements (Figure 2 and Figure 3), the frequencies of circulating total B-cells with a memory phenotype (CD27^+^) and IgA isotype were negatively correlated with plasma RBD IgG Ab levels, while the frequency of total and IgM^+^Tet^++^ B-cells was negatively correlated with the frequency of total Tet^++^ B-cells (Appendix A). In contrast, the frequency of IgA^+^Tet^++^ B-cells positively correlated with the frequency of total Tet^++^ B-cells (Appendix A). For readouts of CD4^+^ and CD8^+^ T-cell proliferation in vitro (Figure 4), none significantly predicted the three major SARS-CoV-2 immunity outcomes (Appendix A). Finally, for readouts of plasma cytokines (Appendix A), there was a tendency for a negative correlation between IL-6 levels and plasma RBD IgG and IgA levels and the frequency of Tet^++^ B-cells, reaching statistical significance only for *p*-values, upon adjustment for numerical and/or categorical variables (Appendix A). Additionally, plasma IL-13 levels were negatively correlated with RBD IgA levels and with the frequency of Tet^++^ B-cells, with *p*-values reaching statistical significance, mainly upon adjustment for numerical and categorical variables (Appendix A). Finally, considering participant recruitment at variable TSI (Appendix A), adjustments performed for TSI revealed TSI-independent positive predictors of plasma RBD IgG Abs (i.e., S1 IgG/IgA, S2 IgG, and NC IgG Abs), RBD IgA Abs (i.e., S1 IgG/IgA and S2 IgG/IgA Abs), circulating Tet^++^ B-cells (i.e., S2 IgG and NC IgA Abs, and IgA+ Tet++ B-cell), as well as negative predictors of plasma RBD IgG Abs (i.e., CD27+ B-cells, IgG+ B-cells, IgA+ B-cells, CD27+ Tet^++^ B-cells) and circulating Tet^++^ B-cells (i.e., B-cells, IgM+ Tet^++^ B-cells, IL-6), as shown in Table 2.

### 3.8. Persistence of Circulating Tet^++^ B-Cells in HNC Patients

To explore the duration of SARS-CoV-2-specific B-cell immunity in HNC patients, the frequency of Tet^++^ B-cells was monitored in a subgroup of HNC participants (n = 25) at visit 1 and visit 2 (median TSI 117 and 341 days, respectively) (Figure 5A). In terms of total CD19^+^ B-cells, a decrease in their frequency was observed at visit 2 versus visit 1, with no changes in the expression of the memory marker CD27, but with an increased frequency of the IgM isotype (Figure 5B,C). Variations in the frequency of circulating Tet^++^ B-cells from visits 1 to 2 were heterogeneous among HNC participants, with no statistically significant differences between the two visits (Figure 5D, left panel). Immunization status was not associated with a particular trend in the frequency of circulating Tet^++^ B-cells. However, a decreased frequency of CD27^+^ Tet^++^ B-cells was observed at visit 2 versus visit 1 (Figure 5D, right panel), without changes in the isotype of Tet^++^ B-cells (Figure 5E). Finally, in this group of HNC participants, TSI was negatively correlated with the frequency of Tet^++^ B-cells at visit 1 but not at visit 2 (Figure 5F). Finally, stratification by immunization status revealed no significant differences in the frequency of circulating Tet^++^ B-cells between visit 1 and visit 2 among the HNC groups (Figure 5G). Taken together, these results indicate the long-term persistence of circulating Tet^++^ B-cells in HNC patients, although with a reduction in the memory phenotype, up to day 717 post-immunization.

## 4. Discussion

In this study, we investigated the amplitude and duration of natural and vaccine-induced SARS-CoV-2 humoral and cellular immunity in a cohort of 49 HNC patients admitted for oncologic treatment at Coltea Hospital, Bucharest, Romania. The enrollment of participants in our study at visit 1 started in August 2021 and finalized in February 2022. The majority of participants were recruited between August and December 2021, when the SARS-CoV-2 Delta variant was predominant in Romania. Plasma samples were used for the measurement of SARS-CoV-2 spike S1, S2, RBD, and NC Abs, while PBMCs were used to monitor RBD-specific B-cells (Tet^++^) ex vivo, and the proliferation of B-cells and CD4^+^ and CD8^+^ T-cells in response to SARS-CoV-2 Spike and NC in vitro. In this cohort of HNC patients, plasma RBD IgG/IgA Abs (with potential neutralizing capacity) and circulating Tet++ B-cells (a likely source of RBD Abs) were identified as major immunological outcomes induced at highest levels upon natural infection, as previously reported [12,13]. Plasma levels of RBD IgG/IgA Abs were positively correlated with the frequency of circulating Tet++ B-cells, with the amplitude of these three outcomes being independent on the time since immunization (TSI), reflecting the persistence of SARS-CoV-2 humoral immunity, at least for the period of time studied (median: 97 days, range 7–315 days post-immunization). The proliferation of B-cells in response to Spike and NC in vitro was the highest in convalescent HNC patients and positively correlated with plasma NC IgG Ab levels, but not with the frequency of circulating Tet++ B-cells, indicating a disconnect between these two B-cell readouts, likely explained by proliferation deficits of Tet++ B-cells. Nevertheless, Tet++ B-cell frequencies were proportional to the proliferation of CD4^+^ T-cells in response to Spike protein. The value of “hybrid immunity” [57,58,59] was observed in this HNC cohort in terms of SARS-CoV-2 RBD IgG/IgA Abs levels that were higher in hybrid immunity compared with vaccinated participants. In addition, levels of SARS-CoV-2 RBD IgG/IgA Abs strongly positively correlated with levels of NC IgG/IgA Abs, indicative of the beneficial effects of combined vaccine and natural exposure SARS-CoV-2 immunity. Finally, our results revealed the ability of HNC patients to develop an effective and long-lasting SARS-CoV-2 immunity, as reflected by plasma SARS-CoV-2 Abs and circulating Tet^++^ B-cells at levels comparable with controls, as well as the persistence of circulating Tet^++^ B-cells between two subsequent visits at a median TSI of 117 and 341 days. These results have clinical relevance for the management of patients with HNC in the context of the persistent circulation of SARS-CoV-2 VOCs. These findings are also informative in the context of the most recent clinical trials conducted by Moderna using personalized mRNA vaccines to promote anticancer responses in people with solid cancers [41].

The fact that plasma levels of SARS-CoV-2 RBD IgG/IgA Abs and the frequency of circulating RBD-specific B-cells were similar between controls and HNC is intriguing in the context where cancer patients are considered at high risk for COVID-19 [15,60,61]. Nevertheless, evidence exists that patients with hematologic, but not solid, tumors exhibit a decreased ability to mount effective humoral responses against SARS-CoV-2 when compared with non-oncologic participants [28]. Our cohort included vaccinated and unvaccinated HNC participants who were naturally exposed to SARS-CoV-2 and developed mild (i.e., self-controlled symptoms that lasted less than 14 days, including mild fever, myalgia, headache, fatigue, and respiratory symptoms without dyspnea, pneumonia, or need for oxygen, without hospitalization) to moderate (i.e., hospitalized for persistent dyspnea after 14 days, with radiological evidence of COVID-19 pneumonia and need for oxygen supplementation and treatment) symptoms. Severe SARS-CoV-2 infection symptoms, defined as COVID-19 pneumonia-associated severe dyspnea requiring intensive care unit admission for mechanical ventilation and treatment, were not reported by the HNC participants in our study. HNC patients detected as positive for SARS-CoV-2 infection by RT-PCR received prompt treatment using the available antiviral, anticoagulant, and anti-inflammatory drugs, at home or upon hospitalization, and were followed up with by the physicians until the resolution of symptoms. Despite the heavy oncologic treatments in HNC patients, the possibility that HNC pathology is associated with natural resistance to severe COVID-19 has been reported in the literature. Indeed, previous studies have reported decreased expression of SARS-CoV-2 receptors/co-receptors in the tumor tissues of HNC patients [62,63,64,65,66]. In line with this possibility, a recent study provided evidence that HNC patients have a genetic predisposition for a mild form of COVID-19 [67]. Moreover, a very recent Mendelian randomization study found no link between genes identified in Gene-Wide Association Studies (GWAS) and susceptibility to COVID-19 and HNC and suggested a positive effect of viral infections on cancer outcome via mechanisms dependent on the type I interferon alpha receptor 2 (IFNAR2) [68]. It is also noteworthy that one study performed in Denmark concluded that HNC patients were not at high risk of acquiring SARS-CoV-2 infection [69]. Although evidence exists in the literature on the common mechanisms involved in antiviral immunity and oncogenesis (e.g., p53 [70]), further studies are needed to molecularly characterize host factors involved in these processes at the respiratory epithelial barrier in the context of SARS-CoV-2 infection and HNC pathology.

One way to distinguish between convalescent and vaccinated individuals is the detection of humoral/cellular immune responses against SARS-CoV-2 NC, which is induced only upon natural infection [10,12,13]. Consistently, in our cohort, the detection of NC Abs, as well as B-cells and CD4^+^ and CD8^+^ T-cells proliferating in response to recombinant SARS-CoV-2 NC proteins, was observed only in convalescent participants. Multiple studies have demonstrated the superiority of hybrid immunity induced by vaccination before and after natural infection in terms of amplitude, breath, and duration of protective SARS-CoV-2-specific immune responses [8,9,57,58,59,71]. In our study, plasma SARS-CoV-2 Abs directed against Spike S1, S2, and RBD proteins of IgG and IgA isotypes were similarly high in convalescent and hybrid immunity but were significantly higher compared with vaccinated HNC patients. Similarly, the frequency of circulating Tet^++^ B-cells and the proliferation (CFSE^low^) of B-cells and CD4^+^/CD8^+^ T-cells in response to recombinant Spike proteins were the most robust in convalescent participants compared with vaccinated participants. Compared with total B-cells, Tet^++^ B-cells were enriched in IgG and IgA isotypes, although the IgM isotype remained predominant, with no differences in the Ig isotypes between vaccinated, convalescent, and hybrid immunity in the HNC group. Of note, there was a significant positive correlation between the frequency of Tet^++^ B-cells and plasma levels of RBD Abs of both IgG and IgA isotypes, suggesting that circulating Tet^++^ B-cells represent the source of these RBD Abs. In addition, the proliferation of CD4^+^ T-cells in response to Spike proteins was positively correlated with the frequency of Tet^++^ B-cells, in line with the importance of CD4^+^ T-cells (e.g., follicular helper) in sustaining B-cell proliferation [8,9].

Protection against infection is ensured by Abs that neutralize SARS-CoV-2 entry into host cells [2,5]. SARS-CoV-2 Abs directed against RBD typically exhibit a neutralizing capacity [72]. Therefore, among the multitude of immune parameters monitored in this study, plasma levels of RBD-specific Abs of IgG and IgA isotypes and the frequency of circulating RBD-specific Tet++ B-cells were considered the major immune outcomes. A linear regression model was used to identify immunological predictors of these immune outcomes. For the RBD IgG Ab outcome, the positive predictors identified included the S1, S2, and NC Abs of IgG isotype only and the frequency of CD19^+^CD27^+^ B-cells and CD19^+^ B-cells with an IgA isotype. For RBD IgA Ab outcome, the positive predictors identified included S1 and S2 Abs of both IgG and IgA isotypes. Finally, for the Tet^++^ B-cell outcome, the positive predictors identified included S2 IgG and NC IgA Abs, as well as the frequency of CD19^+^ B-cells and Tet^++^ B-cells with an IgA isotype, while the frequency of IgM^+^Tet^++^ B-cells was a negative predictor. These predictors remained significant upon adjustments based on numerical (i.e., age, BMI, and TSI) and categorical (i.e., sex, smoking, alcohol, toxic environment, and diabetes) cofounding factors.

Among the panel of 25 cytokines quantified in the plasma, a significant increase in HNC versus control groups was observed only for CCL20, IL-21, and IL-33. Of note, CCL20 was shown to be upregulated in HNC patients, contributing to tumor progression [73]. IL-21 is produced by follicular helper CD4^+^ T-cells that sustain B-cell functions [74], while IL-33 levels are associated with exacerbated pulmonary inflammation in COVID-19 [75]. Among the HNC patients, CCL20, IL-21, and IL-33 levels did not vary relative to immunization status. However, Spearman correlations revealed that plasma levels of IL-6 and, to a lesser extent, IL-10 and IL-13, were negatively correlated with plasma levels of RBD IgG and IgA Abs and the frequency of Tet^++^ B-cells. In addition, plasma levels of IL-6 were identified as moderate negative predictors for the frequency of Tet^++^ B-cells in the linear regression model, with and without adjustment for numerical and categorical confounding factors. This is consistent with the knowledge that IL-6 is a marker of SARS-CoV-2 infection severity and that tocilizumab, an anti-IL-6 Ab, is used for the treatment of COVID-19 pneumonia [61,76,77,78].

Finally, the longitudinal follow-up in a fraction of 25 HNC participants revealed no significant changes in the frequencies of circulating Tet^++^ B-cells at similar frequencies between the two visits at a median TSI of 117 (visit 1) and 341 days (visit 2). Similar to the findings at visit 1, a major fraction of Tet^++^ B-cells exhibited an IgM isotype, while minor fractions presented IgG and IgA isotypes at visit 2. However, a decreased frequency of Tet^++^ B-cells with the CD27^+^ memory phenotype was observed at visit 2 versus visit 1, indicating either a decline in the survival capacity of Tet^++^ B-cells or a more recent activation of these cells, considering the strong likelihood of SARS-CoV-2 re-exposure and/or re-infection in 2022–2023. Nevertheless, the persistence of Tet^++^ B-cells up to 717 days post-immunization (median TSI: 341 days) in this cohort of HNC patients indicates their capacity to mount SARS-CoV-2 humoral immunity, similar to the general population [12,79]. Thus, patients with HNC possess immunological competence to develop durable humoral immune response against SARS-CoV-2 infection. Ongoing surveillance is essential to monitor the longevity of SARS-CoV-2 adaptive immunity in high-risk groups, such as patients with HNC, particularly in the light of virus rapid evolution leading to the continuous emergence of novel VOCs. Immune monitoring should consider not only total antibody levels but also antigen-specific memory B-cells, which may persist even when antibody titers decline [80,81]. Most recent studies reported on the duration of B-cell responses 2–3 years upon vaccination and/or natural infection in health care workers and groups of risk patients [82,83]. Longitudinal immune profiling can help guide personalized vaccination schedules, identify patients at risk for inadequate protection, and optimize the use of adjunctive therapies, such as long-acting monoclonal antibodies or mucosal vaccines [84,85,86].

This study has several limitations. First, we noted sample size limitations linked to the difficulty of enrolling HNC participants and matched non-oncologic controls during the time of SARS-CoV-2 pandemic. To assess the impact of this limitation, a post hoc sample size calculation was performed with the results on SARS-CoV-2 IgG Abs (Figure 1A) using a two-sample t-test at a type 1 error level of 0.05. The post hoc test indicated that a minimum of n = 13 controls and n = 49 HNC participants was sufficient for achieving a power >80% for detecting a high Cohen’s d size effect (d > 0.8) when comparing controls versus HNC for Spike S1, S2, and RBD Ab levels. In contrast, for SARS-CoV-2 IgA Ab levels (Figure 1C), with the sample size used, the test proved to be too underpowered to allow for valid conclusions on the differences between controls and HNC participants. Thus, our conclusions are valid for SARS-CoV-2 Abs of IgG isotypes; however, for IgA isotypes, an increase in sample size is required. Similarly, due to sample size limitations, we were unable to stratify HNC participants per cancer pathology and type of vaccines, which are aspects that can influence immune responses [87]. Nevertheless, the stratification of patients with HNC based on immunization status showed statistically significant differences between the groups. Post hoc sample size calculations were also performed for the results shown in Figure 1B,D using a parametric one-way ANOVA test at a type 1 error level of 0.05. The results indicated that a minimum of n = 6 and n = 11 per group was sufficient to achieve a power >80% in detecting a significant difference, with a high Cohen’s f2 size effect (f2 > 0.35), in at least one comparison (vaccinated. vs. hybrid, vaccinated vs. convalescent, or hybrid vs. convalescent) for plasma IgG Abs against Spike S1, S2, and RBD, as well as NC in Figure 1B and for plasma IgA Abs against Spike S1, S2, and RBD, as well as NC, in Figure 1D, respectively. In addition, we used a post hoc non-parametric Kruskal–Wallis test, which indicated that optimal sample sizes should be majored by 15%, which gives n = 7 and n = 13 for the results in Figure 1B,D, respectively. Thus, we considered that the sample size used to compare differences between HNC groups stratified into vaccinated (i.e., n = 21), convalescent (i.e., n = 13), and hybrid immunity (i.e., n = 15) was sufficient to support our conclusions. Other limitations are linked to the heterogeneity of patients with HNC in terms of tumor grade and oncologic treatment (Appendix A). In our cohort, we did not observe differences in SARS-CoV-2 immunity outcomes between HNC patients who received (n = 27) or did not receive (n = 21) oncologic treatment at study enrollment (Mann–Whitney *p* > 0.5). Moreover, we acknowledge differences in age between HNC and controls and the underrepresentation of women in our HNC cohort (Table 1), the latter being attributed to the highest prevalence of HNC in males [31,32,33,34]. Furthermore, in our effort to enroll as many patients with HNC as possible, a group of oncologic participants with high frailty linked to invasive surgery and heavy oncologic treatment, the TSI varied at enrollment. Therefore, we studied the correlation between TSI and SARS-CoV-2 immunological outcomes and observed no statistical significance, indicating the stability of immune responses at the TSI studied (range: 7–315 days). Linear regression models adjusted for variables also revealed TSI-independent predictors of major immunological outcomes (Table 2). Another limitation is related to the fact that our immune monitoring studies were confined solely to analyses of peripheral blood samples and may not reflect the situation at mucosal barrier surfaces, especially for IgA Abs that were only measured in plasma. Therefore, we acknowledge the importance of investigating mucosal immunity in HNC participants at the upper and lower respiratory barriers, which are key to SARS-CoV-2 transmission [88] and immunity mounting [89,90]. Also, the neutralization capacity of anti-Spike and anti-RBD Abs was not tested but only predicted based on previous studies. Future studies should explore in parallel the duration of neutralizing SARS-CoV-2 immune responses at both the upper respiratory tract (nasal and salivary secretions) versus in peripheral blood, both of which may influence the interpretation of long-term immunity. Finally, the study did not assess immune responses to SARS-CoV-2 VOCs, nor could it account for possible undetected reinfections during follow-up. Incorporating viral sequencing and serologic evidence of reinfection in future studies will be essential for refining our understanding of immune durability and variant-specific protection in immunocompromised populations.

## 5. Conclusions

In conclusion, our study documents the amplitude and duration of humoral and cellular immunity to SARS-CoV-2 in a cohort of 49 patients with HNC admitted for oncologic treatment at Coltea Hospital, Bucharest, Romania, between August 2021 and March 2022 [40]. Briefly, we report (i) similarities in plasma levels of RBD IgG/IgA Abs and the frequency of circulating Tet^++^ B-cells between HNC and control groups; (ii) a positive correlation between circulating Tet^++^ B-cells and plasma levels of RBD IgG/IgA Abs, regardless of the TSI at the time of study initiation (visit 1; median TSI: 97 days, range: 7–315 days); (iii) the superiority of SARS-CoV-2 immunological responses in convalescent/hybrid immunity versus vaccinated participants, coinciding with the superior SARS-CoV-2 Spike- and NC-specific proliferative capacity of B-cells in this group; and (iv) the long-term persistence of circulating Tet++ B-cells in the peripheral blood of HNC patients upon natural infection and/or vaccination up to a median TSI of 341 days at visit 2 (range 107–717 days). Longitudinal follow-up studies are required to determine whether these major immunological outcomes predict long-term protection against SARS-CoV-2 re-infection in patients with HNC, especially in the context of new VOCs that circulate in the general population. Taken together, our results demonstrate the capacity of HNC patients to mount effective and long-lasting humoral and cellular immune responses to SARS-CoV-2 upon natural infection and/or vaccination, despite immunosuppressive radio/chemotherapy and invasive surgical interventions. These findings open the path for new investigations on the link between antiviral and antitumor immunity in specific oncological populations, mainly in the context of the most recent clinical trials for cancer vaccines (e.g., personalized mRNA vaccines) used to treat solid tumors such as HNC [41,91].

## Figures and Tables

**Figure 1 viruses-17-00848-f001:**
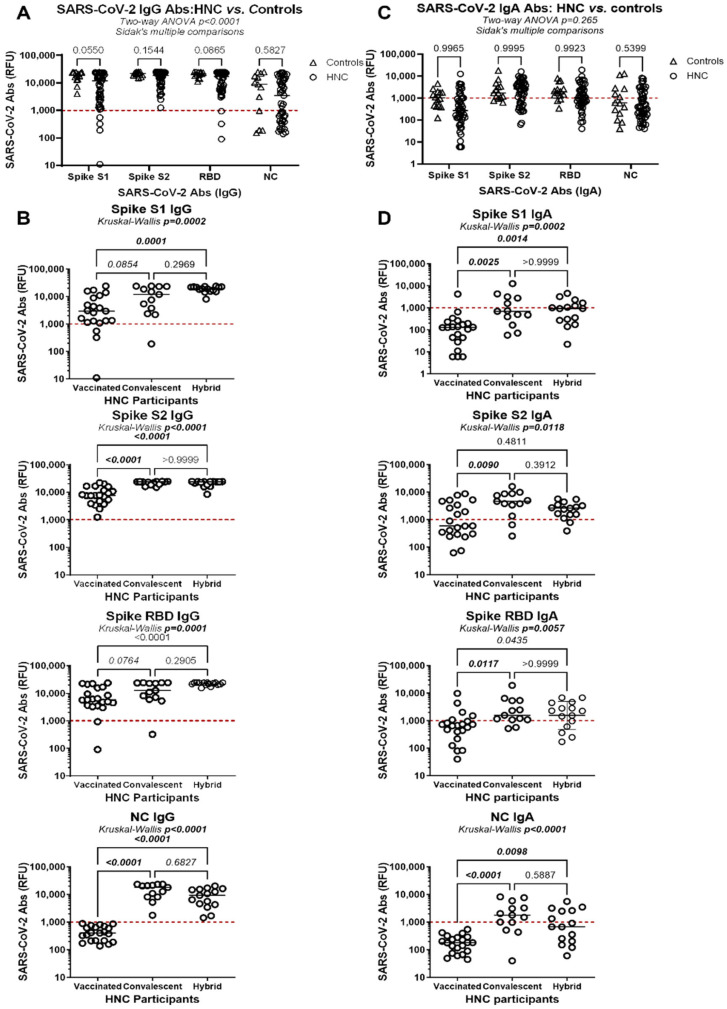
Plasma levels of SARS-CoV-2 IgG and IgA Abs in HNC participants upon vaccination and/or natural infection. Plasma isolated from whole blood served for the quantification of IgG (**A**,**B**) and IgA (**C**,**D**) Abs) against SARS-CoV-2 RBD, S1, S2, and NC using the Luminex xMAP-based multiplex assay. Results are expressed as relative fluorescence units (RFUs). (**A**,**C**) Shown are the levels of SARS-CoV-2 RBD, S1, S2, and NC Abs of IgG (**A**) and IgA (**C**) isotype in the plasma of people with HNC (n = 49) versus healthy controls (n = 14). A two-way ANOVA test followed by Sidak’s multiple comparisons determined differences between groups. (**B**,**D**) Shown are plasma levels of SARS-CoV-2 RBD, S1, S2, and NC Abs of IgG (**B**) and IgA (**D**) isotypes among HNC participants stratified based on the immunization status as vaccinated (n = 21), convalescents (n = 13), and hybrid immunity (n = 15). Statistical significance was determined through the Kruskal–Wallis test and subsequent Dunn’s multiple comparisons, with *p*-values indicated in the graphs.

**Figure 2 viruses-17-00848-f002:**
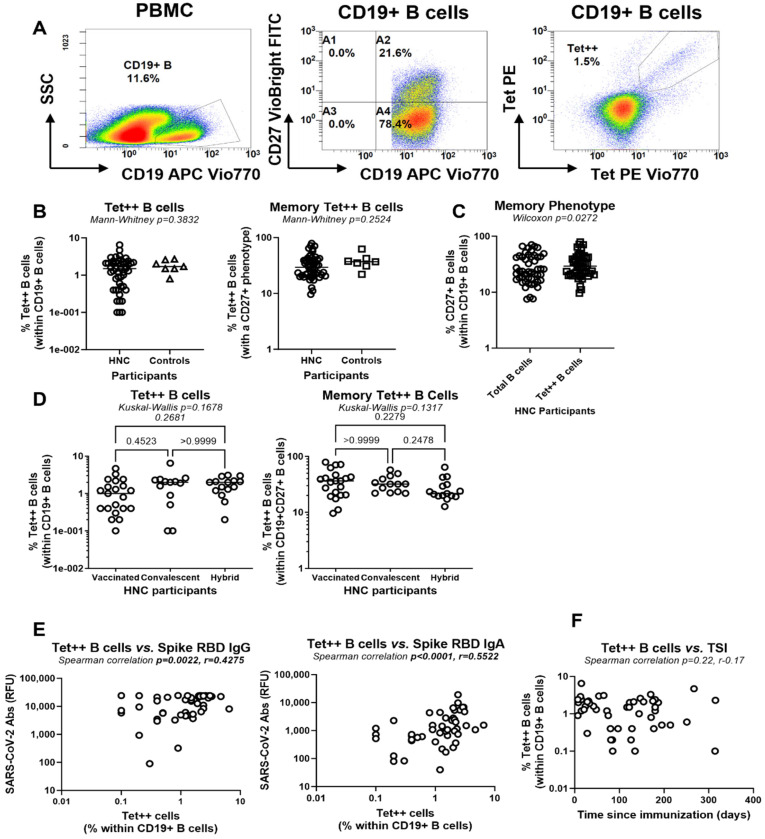
Frequency of RBD-specific B-cells in the blood of HNC participants upon vaccination and/or natural infection. PBMC were isolated from whole blood by Ficoll gradient density centrifugation. (**A**) Shown is the flow cytometry gating strategy in one representative control participant for the identification of a phenotypic characterization of SARS-CoV-2 RBD-specific B-cells within PBMC ex vivo. Tetramers (Tets) were generated by incubating biotin-labeled SARS-CoV-2 RBD protein with streptavidin-PE and streptavidin-PE-Vio770. Surface staining was performed concomitantly with CD19 and CD27 Abs conjugated with the APC Vio770 and FITC fluorochromes, respectively. Lymphocytes were identified based on their size and granularity as indicated by the forward scatter (FSC) and sideward scatter (SSC) signals. B-cells were identified by expression of CD19. Memory B-cells were identified as CD19^+^CD27^+^ cells. SARS-CoV-2 RBD-specific B-cells were identified as cells co-expressing Tet PE and Tet PE-Vio770 (Tet^++^). (**B**–**D**) Shown is the frequency of total (**left panel**) and memory (**right panel**) Tet^++^ B-cells in HNC (n = 49) versus control (n = 7) participants (**B**), the expression of CD27 on total versus Tet^++^ B-cells from HNC participants (**C**) and among HNC participants stratified based on the immunization status in vaccinated (n = 21), convalescent (n = 13) and hybrid immunity (n = 15) (**D**). Mann–Whitney (**B**), Wilcoxon (**C**), and Kruskal–Wallis with Dunn’s multiple comparisons (**D**) determined differences between the two groups. (**E**) Shown are the Spearman correlation *p-* and r-values for the correlation between the frequency of total Tet^++^ B-cells within PBMC and plasma levels of SARS-CoV-2 RBD IgG (**E**, **left panel**) and IgA (**E**, **right panel**) Abs in all HNC participants. (**F**) Shown is the correlation between the frequency of Tet^++^ B-cells and the time since immunization (TSI), with Spearman correlation *p-* and r-values indicated in the graphs.

**Figure 3 viruses-17-00848-f003:**
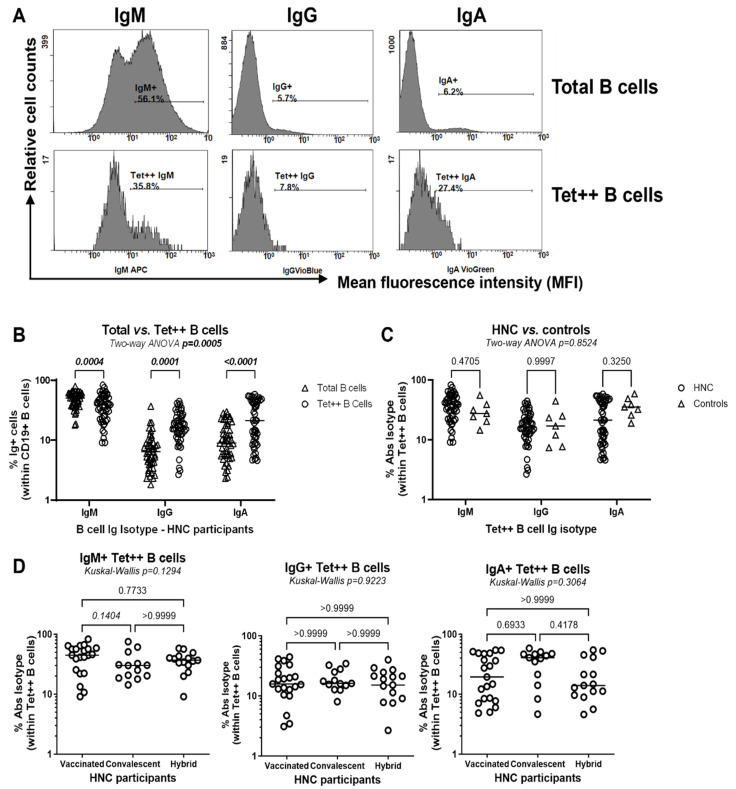
Isotypic characterization of RBD-specific B-cells in the blood of HNC participants upon vaccination and/or natural infection. (**A**) Shown is the gating strategy in one representative control participant for the isotypic characterization (IgM, IgG and IgA) of total and Tet^++^ B-cells, identified as shown in Figure 2A, in control (n = 7) and HNC participants (n = 49). (**B**–**D**) Shown are the isotype of total versus Tet^++^ B-cells within HNC participants (**B**), the isotype of Tet^++^ B-cells in HNC versus control participants (**C**), as well as the IgM (**D**, **left panel**), IgG (**D**, **middle panel**), and IgA (**D**, **right panel**) isotype of Tet^++^ B-cells in HNC participants classified based on their immunization status in vaccinated (n = 21), convalescent (n = 13), and hybrid immunity (n = 15) (**D**). *p*-values for the two-way ANOVA followed by Sidak’s multiple comparisons (**B**,**C**) and Kruskal–Wallis and Dunn’s multiple comparisons (**D**) are indicated in the graphs.

**Figure 4 viruses-17-00848-f004:**
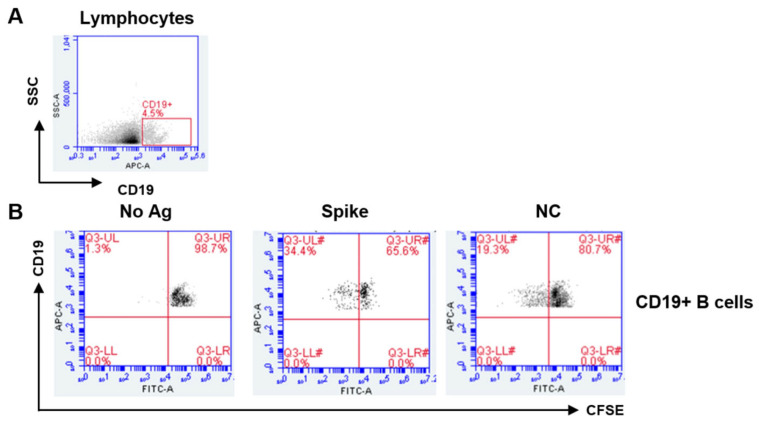
B-cell proliferation in response to SARS-CoV-2 Spike and NC proteins in HNC participants. PBMC were loaded with CFSE, exposed to recombinant SARS-CoV-2 Spike and nucleocapsid (NC) recombinant proteins, and cultured for 6 days. Cells were harvested and stained with APC-conjugated CD19 Abs. The CFSE dilution in cells was used as an indicator of cell proliferation. Shown is the gating strategy used to identify CD19^+^ B-cells (**A**) and their proliferation (CFSE^low^) in response to SARS-CoV-2 Spike and NC antigens (Ag) compared with unstimulated control (No Ag) in one representative HNC participant (**B**). Numbers in the dot plots indicate the percentage of cells identified by manual gating. (**C**) Shown are differences in B-cell proliferation in response to SARS-CoV-2 Spike and NC in HNC participants (n = 35), with the Wilcoxon test *p*-values being indicated in the graph. (**D**) Shown is the proliferation of B-cells in response to SARS-CoV-2 Spike (**D**, **left panel**), and NC (**D**, **right panel**) Ags in HNC participants classified based on their immunization status in vaccinated (n = 13), convalescent (n = 11), and hybrid immunity (n = 11). Kruskal–Wallis and Dunn’s multiple comparison *p*-values are indicated in the graphs. (**E**) Shown are the correlations between the frequency of Tet++ B-cells in the blood and B-cell proliferation (CFSE^low^) in response to SARS-CoV-2 Spike (**E**, **left panel**) and NC (**E**, **right panel**) Ags. (**F**) Shown are the correlations between the plasma levels of NC IgG Abs and B-cell proliferation (CFSE^low^) in response to SARS-CoV-2 Spike (**F**, **left panel**) and NC (**F**, **right panel**) Ags. Spearman correlation *p-* and r-values are indicated in the graphs.

**Figure 5 viruses-17-00848-f005:**
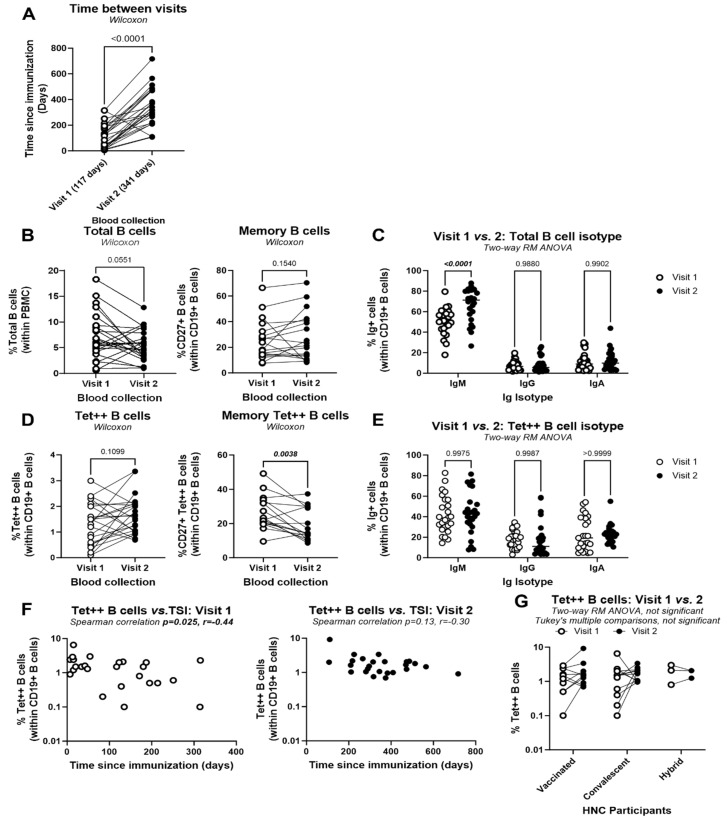
Long-term persistence of SARS-CoV-2 RBD-specific B-cells in the blood of HNC participants. The frequency of total and Tet^++^ B-cells with a memory CD27^+^ phenotype and IgM, IgG, or IgA isotypes, identified by flow cytometry as detailed in the Figure 2 legend, was analyzed at two distinct visits in a group of n = 25 HNC participants. (**A**) Shown is the time since immunization at visit 1 and visit 2 (median time since immunization 117 and 341 days, respectively, as indicated in parentheses). (**B**,**C**) Shown are the frequencies of total (**B**, **left panel**) and memory (CD27^+^) (**B**, **right panel**) CD19^+^ B-cells, as well as their isotype (**C**), at visit 1 compared with visit 2. (**D**,**E**) Shown are the frequencies of total (**D**, **left panel**) and memory (CD27^+^) (**D**, **right panel**) Tet^++^ CD19^+^ B-cells, as well as their isotype (**E**), were compared between visit 1 and 2. Wilcoxon *p*-values (**A**,**B**,**D**) and **a** two-way ANOVA test followed by Sidak’s multiple comparisons (**C**,**E**) determined differences between groups. (**F**) Shown is the correlation between the frequency of Tet^++^ B-cells and the time since immunization at visit 1 (**left panel**) and visit 2 (**right panel**), with Spearman correlation *p-* and r-values being indicated in the graphs. (**G**) Comparison of the frequency of Tet^++^ B-cells between visit 1 and visit 2 among the HNC participants.

**Table 1 viruses-17-00848-t001:** Description of the study cohort.

		HNC	Controls	*p*-Values
Number		n = 49	n = 14	-
Sex *		43 males 6 females	5 males 9 females	*p = 0.0002*
Age * [years; median (range)]		65 (46–78)	56 (30–69)	*p = 0.019*
BMI * [Kgs/m^2^; median (range)]		24.34 (16.05–39.44)	27.09 (23.22–32)	*p* = 0.0696
SARS-C0V-2 symptoms *	Mild	12	7	*p* = 0.1482
	Moderate	18	3	
Immunization status ^&^	Vaccinated	21	4	*p* = 0.1261
	Convalescent	13	7	
	Hybrid immunity	15	2	
Time since immunization (TSI) * [Days; median (range)]		97 (7–315)	122.5 (17–360)	*p* = 0.5471
Diabetes ^&^	Yes	5	3	*p* = 0.266
	No	44	11	
Oncologic pathology	All	49	None	-
	Rhinopharyngeal	1	-	-
	Oropharingeal	11	-	-
	Mandibular	1	-	-
	Laryngeal	28	-	-
	Laryngeal/Pelvilingual	1	-	-
	Pelvilingual	1	-	-
	Parotidian	1	-	-
	Sinus	1	-	-
	Thyroid	1	-	-
	Non-Hodgkin lymphoma	1	-	-
	Unknown primary cancer	2	-	-

HNC, head and neck carcinoma; Controls, participants without oncologic diagnosis; BMI, body mass index; N/A, not available. * For numeric non-parametric variables, median (range) is shown, and the Mann–Whitney test was used for comparison between groups. ^&^ For categorical variables, chi-square (3–4 categories) or Fisher’s exact (2 categories) tests were used for comparison between groups.

**Table 2 viruses-17-00848-t002:** Predictors of the main SARS-CoV-2 immunity outcomes revealed in the linear regression model after adjustment to TSI.

Predictors	Immunity Outcomes	Correlations
S1 IgG Abs * S1 IgA Abs * S2 IgG Abs * NC IgG Abs *	RBD IgG Abs	Positive
CD27^+^ B-cells # IgG^+^ B-cells # IgA^+^ B-cells * CD27^+^ Tet++ B-cells *	RBD IgG Abs	Negative
S1 IgG Abs * S1 IgA Abs * S2 IgG Abs * S2 IgA Abs *	RBD IgA Abs	Positive
S1 IgG Abs # S2 IgG Abs * NC IgA Abs * IgA^+^ Tet^++^ B-cells #	Tet^++^ B-cells	Positive
B-cells * IgM^+^Tet^++^ B-cells * IL-6 #	Tet^++^ B-cells	Negative

*: The positive correlations were statistically significant at both *p*-value and adjusted *p*-values (<0.05) after adjustment for time since immunization (TSI) (Appendix A). #: The negative correlations were only significant for *p*-values (<0.05) after adjustment for TSI (Appendix A).

## Data Availability

All data generated are included in Figures, Tables and Appendix A. Anonymized clinical and demographic information is included in Appendix A, consistent with the principles of protection of human subjects, with the exception of data that cannot be shared for ethical, privacy, or security concerns. Access to primary data is available upon request from the lead author, Petronela Ancuta, via email (petronela.ancuta@umontreal.ca).

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
