# Peer review of "Humoral and Cellular Immune Responses to SARS-CoV-2 in Participants with Head and Neck Cancer"

_viruses, 2025, doi:10.3390/v17060848_

Round 1

Reviewer 1 Report

Comments and Suggestions for Authors

This study investigates the magnitude and persistence of humoral and cellular immune responses to SARS-CoV-2 in patients with head and neck cancer (HNC), comparing them to non-oncologic controls. Using plasma and PBMC samples from 49 HNC patients and 14 controls, the authors measured antibody levels (IgG/IgA), SARS-CoV-2-specific B-cell frequencies, and lymphocyte proliferation in response to Spike and Nucleocapsid proteins. Stratifying subjects by vaccination, prior infection, or hybrid immunity, they found that convalescent and hybrid individuals generally had higher immune readouts than those vaccinated alone. Importantly, SARS-CoV-2-specific B-cells (Tet++) persisted for over 11 months, with minimal decay in frequency. The immune responses in HNC patients were comparable to controls, suggesting retained immunocompetence despite oncologic treatment. These findings may inform clinical strategies for managing infection risk and vaccination efficacy in immunocompromised cancer populations, particularly as new SARS-CoV-2 variants emerge.

To improve the manuscript, we recommend enhancing its contemporary relevance by explicitly discussing how findings remain applicable in 2025, particularly in the context of emerging variants, mucosal vaccines, and mRNA cancer immunotherapy. The study should emphasize its implications for immune monitoring in immunocompromised cancer patients. Figures require clearer resolution, larger fonts, and more consistent labeling to improve readability. Data interpretation would benefit from a stronger clinical framing, addressing why hybrid immunity was not superior and considering the impact of oncologic treatments. Limitations should better reflect the lack of mucosal data, absence of variant-specific analyses, and possible reinfections during follow-up. Additionally, the statistical section would be clearer with defined endpoints, effect sizes, and confidence intervals. Finally, the manuscript’s text—especially the Results section—should be streamlined to improve clarity and flow. 

Reviewer 2 Report

Comments and Suggestions for Authors

Thank you for inviting me to review the manuscript by Mărutescu et al., entitled “Humoral and Cellular Immune Responses to SARS-CoV-2 in Participants with Head and Neck Cancer.” The subject is interesting and clinically relevant; however, the sample size is relatively small, with 49 participants included and longitudinal analyses conducted on only 25 individuals with head and neck cancer (HNC).

I have the following comments:

Page 1: Please specify in the Introduction or Methods section which SARS-CoV-2 variant(s) were circulating in Romania during the study period, when samples were collected.

Page 3, lines 140–141: Please clarify the specific otorhinolaryngology pathologies present in the seven non-HNC participants. Were the other seven participants completely healthy?

Page 4, Section 2.3: According to the methods, you used anti-nucleocapsid antibodies to identify individuals with natural infection or hybrid immunity. However, the half-life of anti-N antibodies is relatively short, and relying solely on these antibodies may lead to false-negative results. Please describe how this limitation was addressed in your study.

Please mention in the Methods or Limitations section that this study did not measure the neutralizing activity of anti-Spike and anti-RBD antibodies.

Since the majority of HNC participants had laryngeal cancer, please elaborate on the chemotherapeutic agents or treatments these patients received, or at least describe the standard treatment protocols used at your center, and how such treatments may have influenced immune responses.

Reviewer 3 Report

Comments and Suggestions for Authors

This is a very well designed and presented work about the humoral and cellular immune responses to SARS-COV-2 in patients with Head and Neck Cancer. Perhaps it is of not so wide interest, due to the particular group of patients, but the authors explain its significance for the rest of immunocompromised patients and the study covers almost all aspects of immune response of interest, that makes it significant among similar studies.

The introduction is very good and it covers all important aspects. In the methods section the centrifuge velocity should be expressed as g and not as rpm, because rpm are meaningful only for this particular centrifuge. On line 178 staining and not straining is probably the correct word. In the results section fig 4B the cursor of the x-axis is not exactly at the same place in No Ag as in spike and NC. The authors have probably a reason, and it is a slight difference but perhaps they could explain why they moved it. On line 467 a decimal point is missing.

The discussion is quite good including the limitations of the study and the authors even give a reason why they find less memory B cells (unexpected) at follow-up.

Round 2

Reviewer 1 Report

Comments and Suggestions for Authors

thanks for having addressed my comments

Reviewer 2 Report

Comments and Suggestions for Authors

Thank you for the revisions. I have no further comments.